# From trivalent to quadrivalent influenza vaccines: Public health and economic burden for different immunization strategies in Spain

Pascal Crépey[1]*, Esther Redondo[2], Javier Díez-Domingo[3], Raúl Ortiz de Lejarazu[4], Federico Martinón-Torres[5,6], Ángel Gil de Miguel[7], Juan Luis López-Belmonte[8‡], Fabián P. Alvarez[9‡], Hélène Bricout[10‡], Míriam Solozabal[11]

1 Department of Quantitative Methods in Public Health, UPRES-EA-7449 Reperes, EHESP, University of Rennes, Rennes, France, 2 Centro de Salud Internacional Madrid Salud, Ayuntamiento de Madrid, Madrid, Spain, 3 Fundación para el Fomento de la Investigación Sanitaria y Biomédica de la Comunitat Valenciana (FISABIO), Valencia, Spain, 4 Centro Nacional de Gripe de Valladolid, Hospital Clínico Universitario de Valladolid, Valladolid, Spain, 5 Servicio Pediatría, Hospital Clínico Universitario de Santiago, Santiago de Compostela, Spain, 6 Grupo de Genética, Infecciones y Vacunas en Pediatría (GENVIP), Instituto de Investigación Sanitaria de Santiago, Universidad de Santiago de Compostela, Santiago de Compostela, Spain, 7 Departamento de Medicina Preventiva y Salud Pública, Universidad Rey Juan Carlos, Madrid, Spain, 8 Sanofi Pasteur España, Madrid, Spain, 9 Sanofi Pasteur Global, Lyon, France, 10 Sanofi Pasteur Europe, Lyon, France, 11 IQVIA, Madrid, Spain

☯ These authors contributed equally to this work.
‡ These authors also contributed equally to this work.
* pascal.crepey@ehesp.fr

**Data Availability Statement:** All relevant data are within the paper.

## Abstract

### Purpose

Quadrivalent influenza vaccine (QIV) includes the same strains as trivalent influenza vaccine (TIV) plus an additional B strain of the other B lineage. The aim of the study was to analyse the public health and economic impact of replacing TIV with QIV in different scenarios in Spain.

### Methods

A dynamic transmission model was developed to estimate the number of influenza B cases prevented under TIV and QIV strategies (<65 years (high risk) and ≥65 years). This model considers cross-protective immunity induced by different lineages of influenza B. The output of the transmission model was used as input for a decision tree model that estimated the economic impact of switching TIV to QIV. The models were populated with Spanish data whenever possible. Deterministic univariate and probabilistic multivariate sensitivity analyses were performed.

### Results

Replacing TIV with QIV in all eligible patients with current vaccine coverage in Spain may have prevented 138,707 influenza B cases per season and, therefore avoided 10,748 outpatient visits, 3,179 hospitalizations and 192 deaths. The replacement could save €532,768

**Funding:** This study was supported by Sanofi Pasteur. IQVIA was funded by Sanofi Pasteur for data collection and preparation of the manuscript. MS is an IQVIA employee. The funder provided support in the form of salaries for authors JLLB, FPA and HB. They played a role in the study design, data collection, and manuscript preparation.

**Competing interests:** I have read the journal's policy and the authors of this manuscript have the following competing interests: PC has received scientific consultancy from Sanofi Pasteur. ER has received funding for scientific consultancy and speaker fees, as well as congress attendance grants from Sanofi Pasteur, GlaxoSmithKline, Merck Sharp and Dohme and Pfizer. JD has received funding for scientific consultancy and speaker fees from Sanofi Pasteur and Seqirus. RO has received funding for scientific consultancy and speaker fees from Sanofi Pasteur, GlaxoSmithKline and Seqirus. FM and his institution have received funding for consultancy and research and speaker fees from Sanofi Pasteur, GlaxoSmithKline, Merck Sharp and Dohme, Pfizer, Astrazeneca, Janssen, Seqirus and Ablynx. AG has received funding for scientific consultancy and speaker fees from Sanofi Pasteur, GlaxoSmithKline, Merck Sharp and Dohme y Pfizer. JLLB, FPA and HB are employed by the commercial Sanofi Pasteur. MS is employed by the commercial IQVIA and works in consultancy projects with Sanofi Pasteur and other Pharmaceutical Companies. JLLB, FPA, HB and MS were involved in the design of the study, data collection and the decision to publish the results. The manuscript was reviewed by the study sponsor prior to submission. This does not alter our adherence to PLOS ONE policies on sharing data and materials. There are no patents, products in development or marketed products to declare.

in outpatient visit costs, €13 million in hospitalization costs, and €3 million in costs of influenza-related deaths per year. An additional €5 million costs associated with productivity loss could be saved per year, from the societal perspective. The budget impact from societal perspective would be €6.5 million, and the incremental cost-effectiveness ratio (ICER) €1,527 per quality-adjusted life year (QALY). Sensitivity analyses showed robust results. In additional scenarios, QIV also showed an impact at public health level reducing influenza B related cases, outpatient visits, hospitalizations and deaths.

## Conclusions

Our results show public health and economic benefits for influenza prevention with QIV. It would be an efficient intervention for the Spanish National Health Service with major health benefits especially in the population ≥65-year.

## Introduction

Influenza is an infectious viral illness that occurs in seasonal epidemics every year [1]. It is mainly caused by influenza type A virus (A/H1N1 and A/H3N2 subtypes) and type B (B/Victoria and B/Yamagata lineages), or any combination of these. Influenza B is more stable than influenza A, with less antigenic drift and consequent immunological stability, although genetic distance of influenza B lineages has been increasing over the years [2].

At global level, the annual incidence rate of influenza is 5–10% in adults and 20–30% in children [1]. In Spain, influenza also constitutes a substantial clinical and socioeconomic burden for society. According to data of the *Instituto de Salud Carlos III* (ISCIII), there are around 600,000 confirmed cases of influenza in adults and 300,000 in children annually [3]. It has been estimated that influenza causes 1.3 million medical consultations every year, more than 140,000 emergency visits [4], 51,000 hospitalizations [5] and between 8,000 and 14,000 deaths in Spain [6, 7]. The total impact of seasonal influenza in Spain costs up to 145–1,000 million euros per year [8]. Hospitalization cost is the key driver, whereas for outpatient care indirect costs, due to absenteeism in the workplace, may be 3.5 times higher than direct costs.

The cases attributable to influenza B virus are distributed across all age groups, particularly in children and young adults [9]. Furthermore, since 2001, surveillance data documented the co-circulation of the two lineages [10, 11] and the unpredictability of predominant B lineage in each season. Whereas in Europe, B strains represent on average 20–25% of all circulating strains [12, 13], in Spain, the circulation of B virus has reached 27.6% between 2000–2001 and 2015–2016 seasons [3, 14]. Another recently published study has confirmed this result, showing that the median proportion of influenza B cases between 2007 and 2017 was 27.2% [15].

Annual vaccination is the most effective way to prevent influenza infections. In 2003 the World Health Organization (WHO) recommended a 75% vaccination coverage rate (VCR) for the older age groups, although by 2010, this objective has not been reached yet in Spain. Specifically, VCR in ≥65-year-old population was 55.7% in the 2017–2018 season [16]. Even more, VCR of healthcare professionals, considered as a group with higher risk for spreading the disease, was only 31% [17].

Twice each year, the WHO recommends the strains of influenza viruses that should be included in the influenza vaccine for the following epidemic season. Quadrivalent influenza vaccine (QIV), which includes two influenza A subtypes (H1N1 and H3N2) and two influenza

B lineages (B/Victoria and B/Yamagata), have been included in these recommendations since the 2013–2014 season, together with the trivalent influenza vaccine (TIV) [18]. In the 2018–2019 season, for the first time the WHO defined the recommendation of QIV as first option [18]. In addition, since 2017 the European Centre for Disease Prevention and Control (ECDC) recommends QIV for influenza prevention [19]. Other European countries have also recognized the value of QIV [20–22]. In Spain, between 2007 and 2017, when the WHO still recommended TIV [18], there has been a mismatch between the circulating B lineage and the one present in the vaccine, in four out of the ten seasons [15]. Overall, an estimated 53.9% of influenza B samples during this period were the influenza B strain not included in the vaccine [3]. As a result, vaccine effectiveness of TIV was reduced [23], which illustrates the need of an influenza vaccine with broader protection against B lineages.

Economic evaluation of health interventions is a useful tool to inform the decision-making process in a resource constrained health system [24]. A previous research on five European countries estimated on the basis of a static health economic model, that the use of QIV instead of TIV throughout 10 seasons (from 2002–2003 to 2012–2013, 2009–10 pandemic excluded) could have prevented up to 150,964 cases of influenza, 13,181 primary care visits, 4,042 hospitalizations and 1,511 deaths, as well as an absenteeism of 18,546 workdays. Altogether, more than 24 million € (21 million € of direct costs and 3 million € of indirect costs) could have been saved [25]. For communicable infectious diseases like influenza, as vaccination may also impact disease transmission, a dynamic model provides a more realistic simulation of disease transmission, capturing changes in the probability of infection over time [26–29]. Hence, dynamic modeling represents the most appropriate approach to quantify the health and economic impact of QIV against seasonal influenza [26, 30]. The aim of this study was to understand and quantify retrospectively the potential public health and economic impact of influenza vaccination with QIV compared to TIV (current standard of care), between 2011 and 2018 seasons in Spain, according to different vaccination scenarios.

## Materials and methods

### Overview

To estimate age-stratified numbers of symptomatic influenza B cases under TIV and QIV strategies in Spain, we adapted a dynamic transmission model developed by *Crépey et al. 2015* previously used in the United States (US) setting [31, 32].

### Model structure

The model is a variation on the compartmental SEIR epidemiological model: susceptible to infection (S), exposed but not infectious (E), infectious (I), and recovered (R) (and therefore immune for a certain time period). In the present model a vaccination compartment was added to include individuals protected by the vaccine. The model considered B virus cross-protection, which accounts for people vaccinated or infected with one B lineage that are partially protected against the opposite B lineage. It also uses a population contact matrix based on data from Italy [33], as no contact matrix directly estimated on Spanish population was available at the time of the study, and population contact characteristics are assumed to be similar in both countries. We considered a latent period of one day and a contagious period of 4.8 days [34]. We assumed that the vaccination campaign was completed ahead of the start of the epidemic with a progressively increasing coverage starting on week 44 and ending on week 50 each year. Asymptomatic infections are also accounted for and assumed to represent a third of all infections [34].

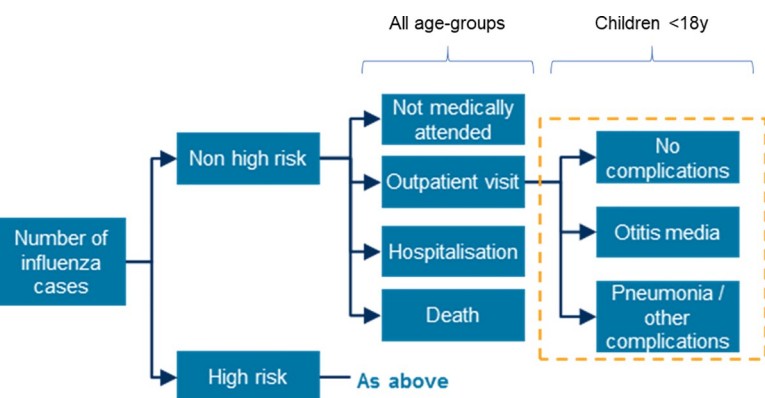

**Fig 1. Flow diagram of the economic decision tree model.**

The economic model is an age-structured decision tree model developed in Excel (Fig 1). The age-stratified symptomatic influenza cases (output of the dynamic model described above) were used as inputs to the economic model. Based on available economic data, the age groups were re-categorized (0–1 years, 2–4 years, 5–14 years, 15–19 years, 20–49 years, 50–64 years 65–69 years, 70–74 years and 75+ years), using age-distribution data of the Spanish population [35]. Influenza cases were stratified between non-high-risk (NHR) and high-risk (HR) patients, based on the presence of underlying medical conditions [36], and were subsequently divided into four categories based on medical outcomes (no medical attention, outpatient visit, hospital admission, and death), with different complication probabilities for each category [37].

## Model calibration

Probabilities of influenza infection–$b_{h1}$ and $b_{h3}$, respectively for A/H1N1 and A/H3N2 and $b_v$ and $b_y$, respectively for B/Victoria and B/Yamagata, were calibrated two by two simultaneously (A together and B together, but not A and B together) for all seasons (2011 to 2018) (Fig 2). In addition, to replicate yearly variations in influenza peaks and dominance of one lineage or subtype over the other, first it was calibrated, for example, $b_{vi}$ and $b_{yi}$ for the first year i = 2011.

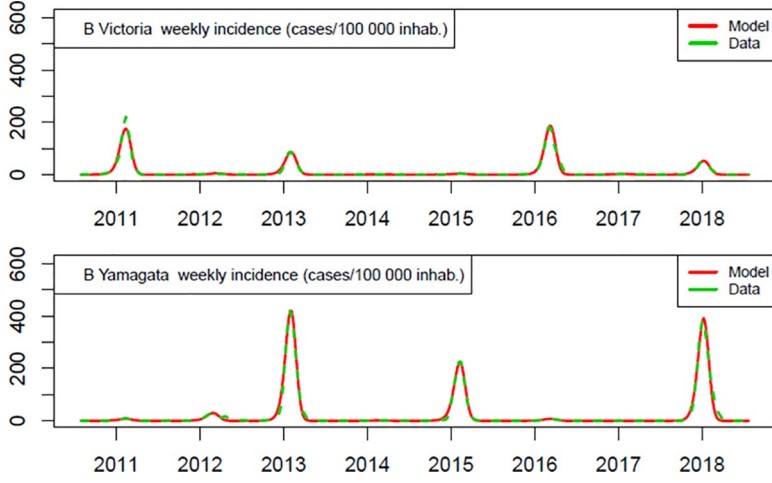

**Fig 2. Calibration results for B Victoria and B Yamagata.**

**Table 1. Model inputs.**

| Variable | Base case | Reference |
|---|:---:|:---:|
| *Vaccine efficacy per strain* | | [38] and CDC unpublished data |
| AH1N1 | | |
| 0–0.5 y | 0 | |
| 0.5–5 y | 0.5085 | |
| 5–10 y | 0.473 | |
| 10–15 y | 0.41 | |
| 15–20 y | 0.41 | |
| 20–40 y | 0.4165 | |
| 40–60 y | 0.6665 | |
| 60–100 y | 0.5 | |
| AH3N2 | | |
| 0–0.5 y | 0 | |
| 0.5–5 y | 0.5085 | |
| 5–10 y | 0.473 | |
| 10–15 y | 0.41 | |
| 15–20 y | 0.41 | |
| 20–40 y | 0.4165 | |
| 40–60 y | 0.6665 | |
| 60–100 y | 0.5 | |
| B Victoria | | |
| 0–0.5 y | 0 | |
| 0.5–5 y | 0.6102 | |
| 5–10 y | 0.5676 | |
| 10–15 y | 0.492 | |
| 15–20 y | 0.492 | |
| 20–40 y | 0.4998 | |
| 40–60 y | 0.7998 | |
| 60–100 y | 0.6 | |
| B Yamagata | | |
| 0–0.5 y | 0 | |
| 0.5–5 y | 0.6102 | |
| 5–10 y | 0.5676 | |
| 10–15 y | 0.492 | |
| 15–20 y | 0.492 | |
| 20–40 y | 0.4998 | |
| 40–60 y | 0.7998 | |
| 60–100 y | 0.6 | |
| Vaccine cross-protection ratio (B strains) (%) | 70 | Estimated from [38] |
| Vaccine coverage (%) | | |
| 0–4 y | 1.68 | [39] |
| 5–14 y | 1.68 | |
| 15–44 y | 5.22 | [40] |
| 45–64 y | 15.67 | |
| 65+ y | 58.16 | |
| Proportion of HR individuals (%) | | |

(*Continued*)

**Table 1.** (Continued)

| Variable | Base case | Reference |
|---|---|---|
| 0–1 y | 7.0 | [39] |
| 2–4 y | 7.0 | |
| 5–14 y | 7.0 | |
| 15–19 y | 11.2 | [40] |
| 20–49 y | 12.1 | |
| 50–64 y | 27.3 | |
| 65–69 y | 37.0 | |
| 70–74 y | 44.3 | |
| 75–79 y | 50.1 | |
| 80+ y | 56.1 | |
| Probability of outpatient visit/flu infection (%) | | |
| 0–1 y | 9.60 | [41], [35] |
| 2–4 y | 9.60 | |
| 5–14 y | 13.66 | |
| 15–19 y | 7.52 | |
| 20–49 y | 7.09 | |
| 50–64 y | 8.99 | |
| 65–69 y | 7,92 | |
| 70–74 y | 6,81 | |
| 75–79 y | 5,40 | |
| 80+ y | 4,04 | |
| Probability of otitis media / influenza-related outpatient visit (%) | | |
| 0–6 m | 5.56 | [42] |
| 6–59 m | 5.56 | |
| 5–9 y | 5.56 | |
| 10–14 y | 5.56 | |
| Probability of pneumonia or other complications / influenza-related outpatient visit (%) | | |
| 0–6 m | 14.05 | [43] |
| 6–59 m | 14.05 | |
| 5–9 y | 14.05 | |
| 10–14 y | 14.05 | |
| Probability of hospitalization / flu case (%) | | |
| 0–1 y | 8.12 | [5, 41] |
| 2–4 y | 5.85 | |
| 5–14 y | 0.46 | |
| 15–19 y | 0.15 | |
| 20–49 y | 0.47 | |
| 50–64 y | 1.76 | |
| 65–69 y | 4.65 | |
| 70–74 y | 3.80 | |
| 75–79 y | 5.16 | |
| 80+ y | 3.98 | |
| Probability of death / flu case (%) | | |

(*Continued*)

**Table 1.** (Continued)

| Variable | Base case | Reference |
|---|---|---|
| 0–1 y | 0.49 | [5, 41] |
| 2–4 y | 0.60 | |
| 5–14 y | 1.46 | |
| 15–19 y | 2.74 | |
| 20–49 y | 6.22 | |
| 50–64 y | 7.84 | |
| 65–69 y | 6.74 | |
| 70–74 y | 5.63 | |
| 75–79 y | 6.63 | |
| 80+ y | 6.47 | |
| *Costs parameters (2017, €)* | | |
| Outpatient visit without complication | | |
| All ages | 54.06 | [44] |
| Outpatient visit otitis media | | |
| 0–14 y | 208.98 | [44] |
| Outpatient visit pneumonia or other complications | | |
| 0–14 y | 208.39 | [44] |
| Hospitalization | | |
| 0–4 y | 2,882.31 | [41] |
| 5–14 y | 2,981.36 | |
| 15–44 y | 4,775.83 | |
| 45–64 y | 5,786.75 | |
| 65+ y | 4,485.89 | |
| Medical cost per death | | |
| 0–4 y | 240.12 | [41] |
| 5–14 y | 1,212.72 | |
| 15–44 y | 9,394.25 | |
| 45–64 y | 16,191.23 | |
| 65+ y | 4,795.73 | |
| Medication | | |
| All ages | 22.52 | [45] |
| Lost workdays: Outpatient visit (N days) | | |
| 15–44 y | 9 | [46] |
| 45–64 y | 9 | |
| Lost workdays: Hospitalization (N days) | | |
| 15–44 y | 30.50 | [46] |
| 45–64 y | 30.50 | |
| Daily earnings for productivity losses | | |
| 15–44 y | 84.66 | [47] |
| 45–64 y | 84.66 | |
| Vaccine price | | |
| TIV | 7.15 | [45] |
| QIV | 9.50 | |
| *Health effects* | | |
| Baseline utility | | |
| 0–4 y | 0.9900 | [48] |
| 5–14 y | 0.9900 | |

(*Continued*)

**Table 1.** (Continued)

| Variable | Base case | Reference |
|---|---|---|
| 15–44 y | 0.9683 | [40] |
| 45–64 y | 0.9140 | |
| 65+ y | 0.7769 | |
| QALY loss per inpatient influenza episode | | |
| 0–18 y | 0.031068493 | [49] |
| 19–49 y | 0.034232877 | |
| 50–64 y | 0.033369863 | |
| 65+ y | 0.032219178 | |
| QALY loss per outpatient influenza episode | | |
| 0–18 y | 0.007863014 | [49] |
| 19–49 y | 0.008821918 | |
| 50–64 y | 0.00690411 | |
| 65+ y | 0.006136986 | |
| Life expectancy | | |
| 0–4 y | 81.18 | [50] |
| 5–14 y | 73.78 | |
| 15–44 y | 54.02 | |
| 45–64 y | 30.20 | |
| 65+ y | 9.46 | |
| *Costs and health effects discounted at 3% [51].* | | |

The final state of year i was the initial state of year i + 1. Then it was calibrated $b_{vi} + 1$ and $b_{yi} + 1$ for year i + 1, taking into account the population immune status acquired in previous years. Each year, the population was vaccinated with a TIV containing the B lineage used the respective year.

Other influenza modelling approaches, including static models, compute the expected probability of infection without vaccination. In contrast, this dynamic model approach allows the impact of vaccination to be directly accounted for, as the estimates are computed under a given vaccination coverage, vaccine composition, and vaccine efficacy (in this case, TIV). The calibration uses a Nelder–Mead simplex algorithm with a least square fitness function. The model and calibration process are implemented in R and take approximately 45 minutes to compute on a 2.9 GHz microprocessing quadricore. The calibration results depicted in Fig 2 show the robustness of the dynamic model.

## Vaccination scenarios

In Spain, influenza vaccination campaigns in most Autonomous Communities are focused on the ≥65-year-old population and on the <65-years-old population but at higher risk of complications (HR population). Vaccination scenarios in the present study were defined taking into account both target populations (S1 Fig).

The scenario 1 compared the current vaccination strategy, where only TIV vaccine is used, versus an alternative vaccination strategy, where all eligible population used QIV at current vaccination coverage rates (Table 1). The following additional scenarios were analysed:

• QIV in < 65 years old (HR population) and TIV for ≥ 65 years old, both at current vaccination coverage rates (scenario 2).

- TIV in < 65 years old (HR population) and QIV for ≥ 65 years old, both at current vaccination coverage rates (scenario 3).

## Probabilities

Table 1 lists all relevant probabilities included in the model.

**Health status.** The proportion of adults (≥18 years) and young adults (15 to 17 years) with high risk of being infected by influenza were obtained from the Spanish National Health Survey [40]. For children this information was obtained from González et al. [39].

**Primary care.** The probability of an outpatient visit per influenza case was estimated from the total number of cases in Primary Care visits (International Classification of Primary Care, Second edition code: R80 influenza) [52] and the total population in Spain [35], between 2011 and 2014. An equal probability was assumed for NHR and HR patients due to the lack of a reliable Spanish source. For children, outpatient visits were further divided between uncomplicated and complicated cases, which were defined as cases that present otitis media [42] and pneumonia or other complications [43].

**Hospitalizations and mortality.** Probabilities of hospitalization were estimated from the hospitalized cases of laboratory-confirmed influenza of the *Instituto de Salud Carlos III* report [5], and the value was adjusted to the pre-specified age groups through Minimum Basic Hospital Data Set (MBDS) [41]; the International Classification of Disease-9 (ICD-9) codes for influenza were 487 and 488. Additionally, an equal probability for NHR and HR patients was assumed. The probability of death within hospitalized individuals was also estimated from the *Instituto de Salud Carlos III* report [5] and adjusted to age groups through MBDS [41].

## Costs

As it is recommended in Spain, a discount rate of 3% was used for costs and health outcomes [51].

**Medical costs.** The cost of outpatient visits in adults was obtained from the eSalud database [44], as well as the cost of outpatient visits in children (<18 years), which also include the cost of physician visits for otitis media and the cost of physician visits for pneumonia or other complications [44]. Hospitalization cost and cost per death were obtained by age group from the National Health System hospital admission's registry (ICD-9 487 and 488) [41]. Patients who died during an episode of hospitalization have an extra cost, based on hospital admission's registry (ICD-9 487 and 488) with an *exitus* discharge; the average cost of all those patients was calculated. Medication cost was calculated as the average cost of influenza antiviral products, obtained from the Spanish official webpage BotPlus 2.0 [45].

**Vaccination costs.** The total yearly number of administered vaccinations was calculated by multiplying the age-specific coverage rates with the corresponding population sizes, as already present in the underlying dynamic model. Vaccine prices of TIV and QIV for the public and private markets were obtained from the Spanish official database BotPlus 2.0 [45]. No administration costs were included in this analysis because these were assumed to be equal in both alternatives.

**Indirect costs.** Days of productivity loss due to outpatient visit or hospitalization were only assigned to adults (≥18 years) and were obtained from Galante et al. [46]. We used the friction costs method approach to count productivity losses [53] and labor elasticity adjustment factor [54], as detailed in a previous publication [32]. The friction cost limits productivity losses of long-term absence to a friction period [55], and the labor elasticity adjustment

quantifies the proportion of reduction in effective labor time due to absence. The friction period was set at 40 days [56, 57] and the elasticity of labor was 0.8 [58]. Days of productivity loss were multiplied by daily earnings [59], assumed equal across all age groups.

## Health effects

Baseline utilities for adults (≥18 years) and young adults (15 to 17 years) were obtained from the Spanish National Health Survey [40]. For children these were derived from García et al. [48]. Lost quality-adjusted life years (QALY) due to influenza were calculated from Hollmann et al. [49], based on utility loss for inpatient and outpatient settings. Life expectancy data was extracted from Spanish projected mortality tables 2016–2065 [50] and was used to calculate the number of LYs lost due to influenza-related deaths.

## Vaccine efficacy

Vaccine efficacy against influenza A and B was calculated by strain (A/H1N1, A/H3N2, B/Victoria and B/Yamagata). In all cases, it was estimated on the basis of Díaz-Granados et al. [38] and by age (CDC unpublished data). The same relative efficacy by age was assumed between different strains and between NHR and HR individuals. The model considered cross-protection between B lineages, estimated from mismatched efficacy data of Díaz-Granados et al. [38]. Potential heterosubtypic immunity between A/H1N1 and A/H3N2 has not been documented.

## Outcomes

The epidemiological model produced weekly symptomatic influenza incidence, incidence per age group and per season, number of influenza cases per subtype and lineage for all years of the study period.

The model enables the estimation of the number of influenza cases by strain and by age groups for all defined scenarios (public health impact). Results are displayed annually and in an aggregated way (for 8 influenza seasons). Burden of influenza avoided due to the replacement of TIV with QIV in different scenarios is also estimated.

## Analysis

Analysis were conducted from both payer and societal perspective. The payer perspective considered the costs assumed by healthcare system [51, 60]. In Spain there is a Public Healthcare System which covers all the health services included in the model but antiviral drugs indicated for influenza. Societal perspective included also the costs of antiviral drugs and indirect cost due to productivity losses [51, 60].

## Sensitivity analysis

A deterministic sensitivity analysis was performed to assess the individual impact of each parameter on the results, using 95% confidence intervals of the parameters when available. Results of the deterministic sensitivity analysis were presented in a Tornado diagram. A probabilistic sensitivity analysis (PSA) was performed to assess the robustness of the results using the commonly accepted distributions: Beta for probabilities, Lognormal for costs and Gamma for utilities and lost workdays. A total of 1,000 iterations were calculated and the results were presented as 95% credibility intervals for the main model outputs.

**Table 2. Reduction of influenza B cases achieved by use of QIV vs TIV, by group of age (over the 8 seasons; 2011–2018).**

| Age groups (years) | TIV Current situation | Scenario 1 (QIV in eligible groups at current VCR) New situation | Absolute difference Cases | Absolute difference Rate* | Relative difference | Scenario 2 (QIV in <65y at current VCR, TIV in >=65y at current VCR) New situation | Absolute difference Cases | Absolute difference Rate* | Relative difference | Scenario 3 (QIV in >=65y at current VCR, TIV in <65y at current VCR) New situation | Absolute difference Cases | Absolute difference Rate* | Relative difference |
|---|---|---|---|---|---|---|---|---|---|---|---|---|---|
| 0–1 | 35,478 | 31,138 | -4,340 | -579 | -12.2% | 34,720 | -758 | -101 | -2.1% | 31,578 | -3,900 | -520 | -11.0% |
| 2–4 | 105,386 | 92,614 | -12,772 | -1,110 | -12.1% | 103,194 | -2,192 | -190 | -2.1% | 93,899 | -11,487 | -998 | -10.9% |
| 5–14 | 992,130 | 873,200 | -118,930 | -2,817 | -12.0% | 971,289 | -20,841 | -494 | -2.1% | 885,930 | -106,200 | -2,516 | -10.7% |
| 15–19 | 689,546 | 612,093 | -77,453 | -3,527 | -11.2% | 675,819 | -13,727 | -625 | -2.0% | 621,033 | -68,513 | -3,120 | -9.9% |
| 20–49 | 2,876,028 | 2,513,312 | -362,716 | -2,122 | -12.6% | 2,810,320 | -65,708 | -384 | -2.3% | 2,552,559 | -323,469 | -1,893 | -11.2% |
| 50–64 | 1,022,743 | 865,327 | -157,416 | -1,733 | -15.4% | 981,888 | -40,855 | -450 | -4.0% | 888,675 | -134,068 | -1,476 | -13.1% |
| 65–69 | 195,966 | 148,505 | -47,461 | -1,575 | -24.2% | 190,207 | -5,759 | -191 | -2.9% | 150,659 | -45,307 | -1,504 | -23.1% |
| 70–74 | 277,914 | 214,001 | -63,913 | -2,222 | -23.0% | 270,283 | -7,631 | -265 | -2.7% | 217,052 | -60,862 | -2,115 | -21.9% |
| 75+ | 1,048,624 | 783,971 | -264,653 | -2,355 | -25.2% | 1,019,038 | -29,586 | -263 | -2.8% | 795,750 | -252,874 | -2,250 | -24.1% |
| **Total** | **7,243,815** | **6,134,161** | **-1,109,654** | **-2,150** | **-15.3%** | **7,056,758** | **-187,057** | **-362** | **-2.6%** | **6,237,135** | **-1,006,680** | **-1,950** | **-13.9%** |

QIV: quadrivalent influenza vaccine; TIV: trivalent influenza vaccine; VCR: vaccination coverage rate.

*Rate per 100,000

## Results

A total of 138,707 influenza B cases would have been avoided per season replacing TIV by QIV in all eligible populations (scenario 1) (1,109,654 cases avoided during 8 influenza seasons), with an impact in all age groups (Table 2). Consequently, related to influenza B, 10,748 outpatient visits, 3,179 hospitalizations and 192 deaths would have been avoided per season (85,982, 25,429 and 1,538 during the whole period), a reduction of 15%, 20% and 21%, respectively (Table 3). Regarding discounted costs, the replacement from TIV to QIV would have led to an increase per year of €27 million in vaccine costs (€219 million for 8 seasons); although it would have saved €0.5 million in outpatient visit costs, €13 million in hospitalization costs, and €3 million in costs of influenza-related deaths (€4 million, €107 million and €20 million during all seasons, respectively). An additional €5 million of productivity losses would have been saved from the societal perspective (€37 million for all 8 seasons) (Table 4).

Taking into account the above-mentioned results (scenario 1), the incremental direct costs would have been €11.7 million (payer perspective) and €6.5 million from societal perspective (discounted savings per year). In terms of health-related outcomes, a total of 4,286 QALYs would have been saved per year. This results in an ICER of €2,751 per QALY gained from a payer perspective and €1,527 from a societal perspective. These results were robust as shown in Fig 3 for the deterministic sensitivity analysis and in Table 5 for the PSA.

In both additional scenarios QIV showed reductions of influenza B cases. Specifically, in scenario 2, where <65-year-old patients were vaccinated with QIV, 23,382 cases would have been avoided per season (187,057 during the 8 influenza seasons), which would have led to avoid 1,883 outpatient visits, 447 hospitalizations and 27 deaths (a reduction of 3% in all cases) (Table 2 and Table 3). The replacement to QIV for part of the population would have implied additional €3 million in vaccine costs, whereas €95,591 would have been saved in outpatient visit costs, €2 million in hospitalization costs and €364,087 million in influenza-related deaths costs. From a societal perspective, €1 million of productivity losses would have been saved (Table 4).

**Table 3. Health outcomes of influenza B avoided with the replacement of TIV by QIV in Spain in different scenarios.**

| Outcomes | Yearly average | | | Total over the 8 seasons (2011–2018) | | | |
|---|---|---|---|---|---|---|---|
| | Current situation | New situation | Difference | Current situation | New situation | Difference | Relative difference |
| **Scenario 1:** (QIV in eligible groups at current VCR) | | | | | | | |
| Number of cases | 905,477 | 766,770 | -138,707 | 7,243,815 | 6,134,161 | -1,109,654 | -15.3% |
| Number of outpatient visits | 73,473 | 62,725 | -10,748 | 587,784 | 501,801 | -85,982 | -14.6% |
| Number of hospitalizations | 15,907 | 12,728 | -3,179 | 127,254 | 101,825 | -25,429 | -20.0% |
| Number of deaths | 905 | 713 | -192 | 7,242 | 5,704 | -1,538 | -21.2% |
| **Scenario 2:** (QIV in <65y at current VCR, TIV in ≥65y at current VCR) | | | | | | | |
| Number of cases | 905,477 | 882,095 | -23,382 | 7,243,815 | 7,056,758 | -187,057 | -2.6% |
| Number of outpatient visits | 73,473 | 71,590 | -1,883 | 587,784 | 572,717 | -15,067 | -2.6% |
| Number of hospitalizations | 15,907 | 15,460 | -447 | 127,254 | 123,678 | -3,576 | -2.8% |
| Number of deaths | 905 | 878 | -27 | 7,242 | 7,026 | -215 | -3.0% |
| **Scenario 3:** (QIV in ≥65y at current VCR, TIV in <65y at current VCR) | | | | | | | |
| Number of cases | 905,477 | 779,642 | -125,835 | 7,243,815 | 6,237,135 | -1,006,680 | -13.9% |
| Number of outpatient visits | 73,473 | 63,784 | -9,689 | 587,784 | 510,274 | -77,510 | -13.2% |
| Number of hospitalizations | 15,907 | 12,940 | -2,967 | 127,254 | 103,518 | -23,736 | -18.7% |
| Number of deaths | 905 | 725 | -180 | 7,242 | 5,804 | -1,438 | -19.9% |

In the last scenario, where patients ≥65 years old were vaccinated with QIV, 125,835 cases would have been avoided per influenza season (1,006,680 during the whole period), together with 9,689 outpatient visits, 2,967 hospitalizations and 180 deaths, a reduction of 13%, 19%

**Table 4. Economic impact of influenza B avoided with the replacement of TIV by QIV in Spain in different scenarios.**

| | Yearly average | | Total over the 8 seasons (2011–2018) | |
|---|---|---|---|---|
| **Current situation** (TIV) | | | | |
| Outpatient visits | 3,710,639 € | - | 29,685,115 € | - |
| Hospitalizations | 65,627,471 € | - | 525,019,766 € | - |
| Deaths | 11,888,318 € | - | 95,106,546 € | - |
| Productivity losses | 34,410,751 € | - | 275,286,007 € | - |
| | New situation | Difference | New situation | Difference |
| **Scenario 1** (QIV in eligible groups at current VCR) | | | | |
| Outpatient visits | 3,177,871 € | -532,768 € | 25,422,972 € | -4,262,143 € |
| Hospitalizations | 52,247,448 € | -13,380,022 € | 417,979,588 € | -107,040,178 € |
| Deaths | 9,347,429 € | -2,540,889 € | 74,779,434 € | -20,327,112 € |
| Productivity losses | 29,749,405 € | -4,661,346 € | 237,995,237 € | -37,290,770 € |
| **Scenario 2** (QIV in <65y at current VCR, TIV in > = 65y at current VCR) | | | | |
| Outpatient visits | 3,615,048 € | -95,591 € | 28,920,386 € | -764,729 € |
| Hospitalizations | 63,693,607 € | -1,933,864 € | 509,548,854 € | -15,470,912 € |
| Deaths | 11,524,231 € | -364,087 € | 92,193,851 € | -2,912,695 € |
| Productivity losses | 33,390,188 € | -1,020,563 € | 267,121,503 € | -8,164,504 € |
| **Scenario 3** (QIV in > = 65y at current VCR, TIV in <65y at current VCR) | | | | |
| Outpatient visits | 3,229,757 € | -480,882 € | 25,838,058 € | -3,847,057 € |
| Hospitalizations | 53,118,653 € | -12,508,818 € | 424,949,222 € | -100,070,544 € |
| Deaths | 9,509,338 € | -2,378,980 € | 76,074,704 € | -19,031,842 € |
| Productivity losses | 30,325,479 € | -4,085,272 € | 242,603,834 € | -32,682,173 € |

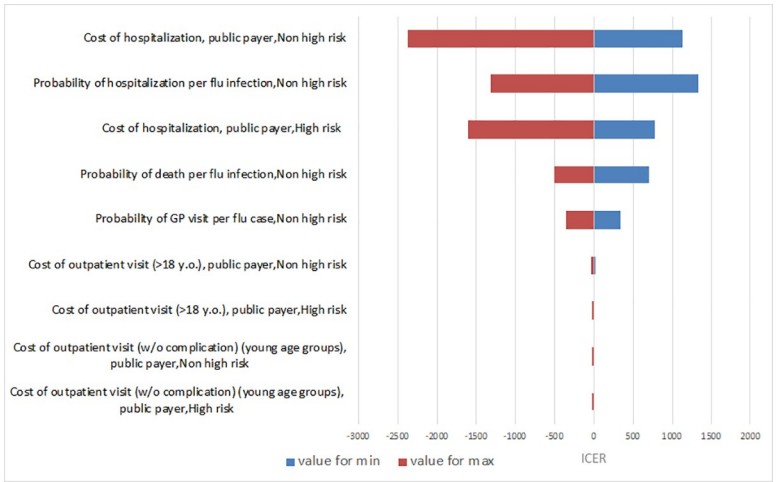

**Fig 3. Deterministic sensitivity analysis on incremental cost-effectiveness ratio in scenario 1.** Parameters were varied within their 95% confidence intervals.

and 20%, respectively (Table 2 and Table 3). Due to the switch from TIV to QIV, vaccination costs would have been incremented by €25 million, although €480,882 would have been saved in outpatient visits, €13 million in hospitalizations and €2 million in influenza-related deaths. €4 million of productivity losses would have been saved additionally from a societal perspective (Table 4).

## Discussion

Our study suggests that the replacement of TIV by QIV in all eligible populations at current vaccination coverage rates would prevent 138,707 influenza B cases per year in Spain. In detail, 23,382 cases would be prevented if the population <65 years switched from TIV to QIV and 125,835 cases if the population ≥65 years was vaccinated with QIV. Consequently, outpatient visits, hospitalizations and deaths would be avoided in all scenarios. Although the complete switch from TIV to QIV to all age groups implies an increase in vaccination cost, the reduction

**Table 5. Results of the probabilistic sensitivity analysis.**

| | Current situation | 95% CI | New situation | 95% CI |
|---|---|---|---|---|
| **Scenario 1: QIV all eligible groups** | | | | |
| TOTAL costs (direct & societal, disc.) (€) | 462,963,000 | [410,462,000; 526,877,000] | 469,429,000 | [420,261,000; 528,359,000] |
| Total number of hospitalizations | 50,630 | [44,800; 57,400] | 47,480 | [42,000; 53,700] |
| Total number of deaths | 2,920 | [2,530; 3,320] | 2,730 | [2,370; 3,100] |
| **Scenario 2: QIV <65y** | | | | |
| TOTAL costs (direct & societal, disc.) (€) | 462,508,000 | [414,574,000; 532,754,000] | 461,826,000 | [414,302,000; 531,372,000] |
| Total number of hospitalizations | 50,790 | [44,900; 57,700] | 50,340 | [44,500; 57,200] |
| Total number of deaths | 2,920 | [2,540; 3,380] | 2,900 | [2,510; 3,350] |
| **Scenario 3: QIV +65y** | | | | |
| TOTAL costs (direct & societal, disc.) (€) | 459,760,000 | [415,006,000; 527,887,000] | 465,658,000 | [423,355,000; 530,595,000] |
| Total number of hospitalizations | 50,880 | [44,900; 56,900] | 47,920 | [42,300; 53,500] |
| Total number of deaths | 2,920 | [2,530; 3,310] | 2,740 | [2,380; 3,110] |

CI: credibility intervals; QIV: quadrivalent influenza vaccine.

of healthcare costs would compensate, showing that QIV is a highly cost-effective alternative both from payer and societal perspective, with an ICER clearly below the usually mentioned Spanish threshold of €25,000 per QALY [61]. Whether budget impact restrictions do not allow complete switching in a single influenza season, population ≥65 years will obtain the greater benefit due to its higher vaccination coverage rates and burden of illness related with hospitalization.

The results of this study show that the population ≥65 years is the population that most benefit from the switch from TIV to QIV. In scenario 1, when all eligible groups were vaccinated with QIV, relative difference (compared with TIV) in avoided cases with ≥65-years and <65 years were 25% and 13%, respectively; and in scenario 3, when only ≥65-year population was vaccinated with QIV, 24% and 11% additional cases were avoided. The higher number of avoided cases led to avoid a higher number of outpatient visits, hospitalizations and deaths in ≥65-year-old patients. These results were as expected, since in Spain, influenza vaccination is recommended to all population ≥65 years and <65 years at high risk. In addition, vaccination coverage of ≥65-year-old population is the highest. Even more, nowadays 10 of the 17 Autonomous Communities have started to include QIV in their influenza vaccination campaigns. *Pérez-Rubio et al.* [8] also published that, although accumulated incidence rate in population >65 years was the lowest, this age group was the most affected in terms of number of influenza-related hospitalizations and deaths.

In Spain, a cost-effectiveness model of QIV versus TIV, based on a static, lifetime, multi-cohort state transition model has been published previously [48]. This analysis, using a one-year scenario, showed that QIV (compared with TIV) would have prevented 18,565 influenza cases, 407 influenza-related hospitalizations and 181 deaths [48]. The reduction of influenza B cases and hospitalizations in our dynamic model was higher than the results of the static model, which can be partially explained by the risk reduction of virus transmission (indirect effect). In both static and dynamic models, the cost of QIV was the same (€9.50) and the cost of TIV was similar (€7.00 and €7.15, respectively), as well as most of the inputs used to populate both models. The costs from the payer perspective were higher with QIV due to a higher vaccine price in both cases; and the difference was offset due to lower indirect costs associated with QIV (societal perspective). Although in both analyses the results showed a QALY gain with QIV, the ICER of QIV over TIV from a societal perspective was €8,748/QALY gained with the static model [48], whereas it was more cost-effective with the dynamic model.

In England, a cost-effectiveness analysis of QIV found that it could be cost-effective for all targeted groups, but especially for children aged 2–11 years [62]. Conversely, population ≥65 year is the most benefited in our analysis. The rationale could be differences in vaccination programs, in England all children attending primary schools are vaccinated, while in Spain only high-risk children are vaccinated and the coverage is low, then herd protection of vaccinating children to protect other age groups would be greater in England than in Spain. Secondly, no GP consultations or hospitalisations for those individuals aged 65 years and older are attributed to influenza B on average each season, while in Spain this age group, as well as children < 5 years, has the higher rates hospitalisations related with influenza.

Until now, the most common approach to economic evaluations of vaccines are cost-effectiveness and cost-utility analyses, although they do not capture all vaccine externalities [63]. Influenza vaccines give indirect protection to non-vaccinated individuals by reducing the transmission of influenza from the vaccinated population. These broader benefits have been increasingly incorporated into economic evaluations [63]. Therefore, the benefits of dynamic models for economic analyses of vaccines have been demonstrated in many studies, such as in *Pradas-Velasco et al* [64]. This Spanish study evaluated the efficiency of seasonal influenza vaccination using both a static and dynamic model approach [64]. Study results showed that

influenza vaccination was not efficient when using static models whereas it was efficient through a dynamic model. This difference was caused by indirect effects on the non-vaccinated population (herd protection) which were not reflected in the static model, although they could be greater than the direct effect [64].

Recently, two guidelines have been published with the objective of providing a consensus on how to apply economic evaluation to infectious diseases vaccines [65, 66]. Both studies recommend that infectious disease models should be dynamic, in order to reflect that vaccination programs can change the infectious disease dynamics. Therefore, vaccination programmes for infectious diseases have an indirect benefit on non-vaccinated individuals by preventing onward transmission in vaccinated people, which cannot be easily captured by static models. Even so, as indicated by the WHO, in some specific circumstances a static approach could be considered; for example when vaccinated groups are unlikely to change population disease transmission substantially (like ≥ 65-year-old population) [67].

This study has limitations given that different assumptions were necessary. Probability of flu complications and QALY loss for influenza B were not available, so we used data for influenza in general, although no significant differences between influenza A and B have been shown in clinical burden [14]. Because inter-individual contact data for the Spanish population were not available, we used an Italian contact matrix [33] assuming that population contact characteristics between Spain and Italy are similar. We also used data related to the natural history of influenza from other countries, however it is widely accepted that influenza natural history is similar among countries. Finally, despite the use of a dynamic model, which represents the recommended method for infectious disease vaccination programs [65], our approach did not capture other vaccination benefits like e.g. the reduction of antimicrobial drugs utilization that reduce antimicrobial resistance [68] or seasonal collapse of hospital services due to influenza outbreaks [1]. These variables would have likely increased the burden of influenza to the healthcare system.

## Conclusions

Using a dynamic model as recommended by most recent vaccine evaluation guidelines, our study shows that QIV could be an efficient intervention for the National Health Service (from a payer perspective), being even more efficient from a societal perspective. This analysis also shows that most health benefits of QIV are obtained replacing TIV in the ≥65-year-old population.

## Supporting information

**S1 Fig. Influenza vaccination strategies analyzed.**
(TIF)

## Author Contributions

**Conceptualization:** Pascal Crépey, Esther Redondo, Javier Díez-Domingo, Raúl Ortiz de Lejarazu, Federico Martinón-Torres, Ángel Gil de Miguel, Juan Luis López-Belmonte, Fabián P. Alvarez, Hélène Bricout.

**Data curation:** Pascal Crépey, Juan Luis López-Belmonte, Fabián P. Alvarez, Hélène Bricout.

**Formal analysis:** Pascal Crépey, Juan Luis López-Belmonte, Fabián P. Alvarez, Hélène Bricout, Míriam Solozabal.

**Resources:** Juan Luis López-Belmonte, Fabián P. Alvarez, Hélène Bricout, Míriam Solozabal.

**Validation:** Pascal Crépey, Esther Redondo, Javier Díez-Domingo, Raúl Ortiz de Lejarazu, Federico Martinón-Torres, Ángel Gil de Miguel, Juan Luis López-Belmonte, Fabián P. Alvarez, Hélène Bricout, Míriam Solozabal.

**Writing – original draft:** Míriam Solozabal.

**Writing – review & editing:** Pascal Crépey, Esther Redondo, Javier Díez-Domingo, Raúl Ortiz de Lejarazu, Federico Martinón-Torres, Ángel Gil de Miguel, Juan Luis López-Belmonte, Fabián P. Alvarez, Hélène Bricout.

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
