## [Decision Letter · Decision Letter 0]

6 Mar 2020

PONE-D-20-03263

From Trivalent to Quadrivalent Influenza Vaccines: Public Health and Economic Burden for Different Immunization Strategies in Spain

PLOS ONE

Dear Dr Crepey,

Thank you for submitting your manuscript to PLOS ONE. After careful consideration, we feel that it has merit but does not fully meet PLOS ONE’s publication criteria as it currently stands. Therefore, we invite you to submit a revised version of the manuscript that addresses the points raised during the review process.

We would appreciate receiving your revised manuscript by 2nd April 2020. To enhance the reproducibility of your results, we recommend that if applicable you deposit your laboratory protocols in protocols.io, where a protocol can be assigned its own identifier (DOI) such that it can be cited independently in the future. For instructions see: http://journals.plos.org/plosone/s/submission-guidelines#loc-laboratory-protocols

We look forward to receiving your revised manuscript.

Kind regards,

Shinya Tsuzuki, MD, MSc

Academic Editor

PLOS ONE

Journal Requirements:

"I have read the journal's policy and the authors of this manuscript have the following competing interests:

PC has received scientific consultancy from Sanofi Pasteur. ER has received funding for scientific consultancy and speaker fees, as well as congress attendance grants from Sanofi Pasteur, GlaxoSmithKline, Merck Sharp and Dohme and Pfizer. JD has received funding for scientific consultancy and speaker fees from Sanofi Pasteur and Seqirus. RO has received funding for scientific consultancy and speaker fees from Sanofi Pasteur, GlaxoSmithKline and Seqirus. FM and his institution have received funding for consultancy and research and speaker fees from Sanofi Pasteur, GlaxoSmithKline, Merck Sharp and Dohme, Pfizer, Astrazeneca, Janssen, Seqirus and Ablynx. AG has received funding for scientific consultancy and speaker fees from Sanofi Pasteur, GlaxoSmithKline, Merck Sharp and Dohme y Pfizer. JLLB, FPA and HB are Sanofi Pasteur employees. MS is an IQVIA employee and works in consultancy projects with other Pharmaceutical Companies.".

We note that one or more of the authors are employed by a commercial company: 'IQVIA, and Sanofi Pasteur'

Additional Editor Comments (if provided):

I believe the manuscript would be of great value for publication, however, both reviewers raised reasonable concerns. Please clarify what they pointed out, especially model structure and parameters, perspective used in economic evaluation, comparison with other countries, and sensitivity analysis about contact matrix.

Reviewers' comments:

Reviewer's Responses to Questions

**Comments to the Author**

1. Is the manuscript technically sound, and do the data support the conclusions?

Reviewer #1: Yes

Reviewer #2: Yes

2. Has the statistical analysis been performed appropriately and rigorously? 

Reviewer #1: Yes

Reviewer #2: Yes

3. Have the authors made all data underlying the findings in their manuscript fully available?

Reviewer #1: Yes

Reviewer #2: No

4. Is the manuscript presented in an intelligible fashion and written in standard English?

Reviewer #1: Yes

Reviewer #2: Yes

5. Review Comments to the Author

Reviewer #1: This is a very interesting and important piece of work for Spanish vaccination policy against seasonal influenza. I highly recommend that this study is published following some minor revisions to the manuscript.

The most important comment that I can make is that the Discussion section fails to mention the context of this analysis by comparing the results presented here to other cost-effectiveness analyses on quadrivalent influenza vaccines in other countries. This analysis found that the QIV would be highly cost-effective for all target populations, whereas other studies (notably one conducted by Public Health England) found the opposite - that is, QIV was cost-effective for the elderly population, but it was less likely to be cost-effective for younger populations, mainly due to the existing QLAIV vaccination programme in schools. Other countries have found different results driven the local epidemiology of influenza B. It would therefore be really helpful to this manuscript if reference could be made to other countries and their CEAs in this domain.

There are several minor comments to make in addition to the main one above:

- Line 12 needs re-writing grammatically. I would suggest "The total impact of seasonal influenza in Spain costs up to 145-1,000 million euros per year

- Line 14 : define indirect costs

- Line 15 : Re-write as "The cases attributable to influenza B..." or "The cases caused by influenza B..."

- Line 24 : Citation 16 is using data that is now 10 years old, and a later estimate of vaccine coverage in the region is presented later in the paragraph. Is citation 16 therefore necessary? Why refer to achieved vaccine coverage of 10 years ago?

Finally, I would also suggest that the authors consider adding to their sensitivity analysis by estimating the impact on their results of the assumption that the Italian contact matrix can be substituted for the Spanish population. Contact patterns can be a key determinant for cost-effectiveness of some vaccination programmes and making assumptions here can be problematic. Even countries that participated in the POLYMOD study still like to perform sensitivity analyses on their contact matrices now and again.

Reviewer #2: 1. Dynamic model

The analysis uses an age-structured SEIR model with an age structure to consider indirect benefits of vaccination. I would like to see more details on the model, and model parameters (i.e., infection rate and recovery rate). Do the authors estimate the model parameters? How vaccination is incorporated in the model? In which parameter does age matters? All of these details are important to understand the procedure and to interpret the results.

2. Perspectives

The authors conduct an analysis from two perspectives: societal and payer perspectives. Please explain what the analysis from each perspective considers. It is common to use the word “perspectives” to describe who pays the cost, but authors seem to use societal perspective to indicate productivity losses. It is confusing and need some clarifications.

3. Comparison across three scenarios

There are three scenarios, and I wonder how the three scenarios are selected in a relation to healthcare policy in Spain. What is the current recommendation and is there any argument to change the recommendation? Also I would like to see some discussion of the results. At the end, which scenario is favored in terms of cost-effectiveness?

6. PLOS authors have the option to publish the peer review history of their article (what does this mean?). If published, this will include your full peer review and any attached files.

Reviewer #1: No

Reviewer #2: No

---

## [Author Response · Author response to Decision Letter 0]

14 Apr 2020

We really appreciate journal, editor and reviewers’ comments which will allow to improve the manuscript quality. Below you can find our response and actions taken to incorporate your comments in the manuscript.

Journal Requirements:

• http://www.journals.plos.org/plosone/s/file?id=wjVg/PLOSOne_formatting_sample_main_body.pdf and

• http://www.journals.plos.org/plosone/s/file?id=ba62/PLOSOne_formatting_sample_title_authors_affiliations.pdf

Response: Done

2. PLOS requires an ORCID iD for the corresponding author in Editorial Manager on papers submitted after December 6th, 2016. Please ensure that you have an ORCID iD and that it is validated in Editorial Manager. To do this, go to ‘Update my Information’ (in the upper left-hand corner of the main menu), and click on the Fetch/Validate link next to the ORCID field.

This will take you to the ORCID site and allow you to create a new iD or authenticate a pre-existing iD in Editorial Manager. Please see the following video for instructions on linking an ORCID iD to your Editorial Manager account: https://www.youtube.com/watch?v=_xcclfuvtxQ

Response: Done

"I have read the journal's policy and the authors of this manuscript have the following competing interests: PC has received scientific consultancy from Sanofi Pasteur. ER has received funding for scientific consultancy and speaker fees, as well as congress attendance grants from Sanofi Pasteur, GlaxoSmithKline, Merck Sharp and Dohme and Pfizer. JD has received funding for scientific consultancy and speaker fees from Sanofi Pasteur and Seqirus. RO has received funding for scientific consultancy and speaker fees from Sanofi Pasteur, GlaxoSmithKline and Seqirus. FM and his institution have received funding for consultancy and research and speaker fees from Sanofi Pasteur, GlaxoSmithKline, Merck Sharp and Dohme, Pfizer, Astrazeneca, Janssen, Seqirus and Ablynx. AG has received funding for scientific consultancy and speaker fees from Sanofi Pasteur, GlaxoSmithKline, Merck Sharp and Dohme y Pfizer. JLLB, FPA and HB are Sanofi Pasteur employees. MS is an IQVIA employee and works in consultancy projects with other Pharmaceutical Companies.".

We note that one or more of the authors are employed by a commercial company: 'IQVIA, and Sanofi Pasteur'

Within your Competing Interests Statement, please confirm that this commercial affiliation does not alter your adherence to all PLOS ONE policies on sharing data and materials by including the following statement: "This does not alter our adherence to PLOS ONE policies on sharing data and materials.” (as detailed online in our guide for authors

http://journals.plos.org/plosone/s/competing-interests) . If this adherence statement is not accurate and there are restrictions on sharing of data and/or materials, please state these. Please note that we cannot proceed with consideration of your article until this information has been declared.

Please know it is PLOS ONE policy for corresponding authors to declare, on behalf of all authors, all potential competing interests for the purposes of transparency. PLOS defines a competing interest as anything that interferes with, or could reasonably be perceived as interfering with, the full and objective presentation, peer review, editorial decision-making, or publication of research or non-research articles submitted to one of the journals. Competing interests can be financial or non-financial, professional, or personal. Competing interests can arise in relationship to an organization or another person.

Please follow this link to our website for more details on competing interests:

http://journals.plos.org/plosone/s/competing-interests

Response: Modified. Since we could not find the adequate field to provide the competing interests statement during the submission process, we include this declaration in the Respond to Reviewers field and the Cover Letter :

I have read the journal's policy and the authors of this manuscript have the following competing interests:

PC has received scientific consultancy from Sanofi Pasteur. ER has received funding for scientific consultancy and speaker fees, as well as congress attendance grants from Sanofi Pasteur, GlaxoSmithKline, Merck Sharp and Dohme and Pfizer. JD has received funding for scientific consultancy and speaker fees from Sanofi Pasteur and Seqirus. RO has received funding for scientific consultancy and speaker fees from Sanofi Pasteur, GlaxoSmithKline and Seqirus. FM and his institution have received funding for consultancy and research and speaker fees from Sanofi Pasteur, GlaxoSmithKline, Merck Sharp and Dohme, Pfizer, Astrazeneca, Janssen, Seqirus and Ablynx. AG has received funding for scientific consultancy and speaker fees from Sanofi Pasteur, GlaxoSmithKline, Merck Sharp and Dohme y Pfizer. JLLB, FPA and HB are employed by the commercial Sanofi Pasteur. MS is employed by the commercial IQVIA and works in consultancy projects with Sanofi Pasteur and other Pharmaceutical Companies. JLLB, FPA, HB and MS were involved in the design of the study, data collection and the decision to publish the results. The manuscript was reviewed by the study sponsor prior to submission.

This does not alter our adherence to PLOS ONE policies on sharing data and materials. There are no patents, products in development or marketed products to declare.

This study was supported by Sanofi Pasteur. IQVIA was funded by Sanofi Pasteur for data collection and preparation of the manuscript. MS is an IQVIA employee. The funder provided support in the form of salaries for authors JLLB, FPA and HB. They played a role in the study design, data collection, and manuscript preparation.

 

Comments to the Author:

3. Have the authors made all data underlying the findings in their manuscript fully available? The PLOS Data policy requires authors to make all data underlying the findings described in their manuscript fully available without restriction, with rare exception (please refer to the Data Availability Statement in the manuscript PDF file). The data should be provided as part of the manuscript or its supporting information, or deposited to a public repository. For example, in addition to summary statistics, the data points behind means, medians and variance measures should be available. If there are restrictions on publicly sharing data—e.g. participant privacy or use of data from a third party—those must be specified.

Response: We are not completely sure what is the additional information required. In the results section when variance measures are estimated they are included, see Table 5. 

Reviewer #1: This is a very interesting and important piece of work for Spanish vaccination policy against seasonal influenza. I highly recommend that this study is published following some minor revisions to the manuscript.

The most important comment that I can make is that the Discussion section fails to mention the context of this analysis by comparing the results presented here to other cost-effectiveness analyses on quadrivalent influenza vaccines in other countries. This analysis found that the QIV would be highly cost-effective for all target populations, whereas other studies (notably one conducted by Public Health England) found the opposite - that is, QIV was cost-effective for the elderly population, but it was less likely to be cost-effective for younger populations, mainly due to the existing QLAIV vaccination programme in schools. Other countries have found different results driven the local epidemiology of influenza B. It would therefore be really helpful to this manuscript if reference could be made to other countries and their CEAs in this domain.

Response: A new paragraph was included in the Discussion section, comparing our results with the study conducted by Public Health England

There are several minor comments to make in addition to the main one above:

- Line 12 needs re-writing grammatically. I would suggest "The total impact of seasonal influenza in Spain costs up to 145- 1,000 million euros per year

Response: Modified

- Line 14 : define indirect costs

Response: Modified

- Line 15 : Re-write as "The cases attributable to influenza B..." or "The cases caused by influenza B..."

Response: Modified

- Line 24 : Citation 16 is using data that is now 10 years old, and a later estimate of vaccine coverage in the region is presented later in the paragraph. Is citation 16 therefore necessary? Why refer to achieved vaccine coverage of 10 years ago?

Response: Modified

Finally, I would also suggest that the authors consider adding to their sensitivity analysis by estimating the impact on their results of the assumption that the Italian contact matrix can be substituted for the Spanish population. Contact patterns can be a key determinant for cost-effectiveness of some vaccination programmes and making assumptions here can be problematic. Even countries that participated in the POLYMOD study still like to perform sensitivity analyses on their contact matrices now and again.

Response: To our knowledge, there is no contact matrix directly estimated on the Spanish population. All European inter-individual matrices display similar features, in particular the assortativeness of contacts among similar age-groups, hence we do not expect any qualitatively significant variations. Nevertheless, we are aware of this potential limitation of the study and we acknowledge it in the discussion. 

Reviewer #2: 1. Dynamic model

The analysis uses an age-structured SEIR model with an age structure to consider indirect benefits of vaccination. I would like to see more details on the model, and model parameters (i.e., infection rate and recovery rate). Do the authors estimate the model parameters? How vaccination is incorporated in the model? In which parameter does age matters? All of these details are important to understand the procedure and to interpret the results.

Response: We agree with the reviewer that the model was not described in large details as it was previously published in another publication. Following the reviewer’s advice, we have added more information regarding the latent and contagious period considered. We have also added information regarding the timing of influenza vaccination (coverage rates by age-group were already in the text). The details of the calibration procedure and its results are already in the method section of the manuscript. However, we chose not to show the estimated probabilities of infection as they are model-dependent and not meaningful for the reader, but they are available upon request. 

2. Perspectives

The authors conduct an analysis from two perspectives: societal and payer perspectives. Please explain what the analysis from each perspective considers. It is common to use the word “perspectives” to describe who pays the cost, but authors seem to use societal perspective to indicate productivity losses. It is confusing and need some clarifications.

Response: A new paragraph explaining perspectives analyses was included in the Methods section.

3. Comparison across three scenarios

There are three scenarios, and I wonder how the three scenarios are selected in a relation to healthcare policy in Spain. What is the current recommendation and is there any argument to change the recommendation? Also I would like to see some discussion of the results. At the end, which scenario is favored in terms of cost-effectiveness?

Response: It is explained in the Vaccination scenarios subsection of Methods. To support it a new figure to be included in Supplementary Material has been developed. In the discussion additional sentences were added regarding different scenarios analyzed

---

## [Decision Letter · Decision Letter 1]

28 Apr 2020

PONE-D-20-03263R1

From Trivalent to Quadrivalent Influenza Vaccines: Public Health and Economic Burden for Different Immunization Strategies in Spain

PLOS ONE

Dear Dr Crepey,

Thank you for submitting your manuscript to PLOS ONE. After careful consideration, we feel that it has merit but does not fully meet PLOS ONE’s publication criteria as it currently stands. Therefore, we invite you to submit a revised version of the manuscript that addresses the points raised during the review process.

I feel that this manuscript is almost ready for publication, however, please insert further references as one reviewer suggested.

We would appreciate receiving your revised manuscript by 11th May 2020. To enhance the reproducibility of your results, we recommend that if applicable you deposit your laboratory protocols in protocols.io, where a protocol can be assigned its own identifier (DOI) such that it can be cited independently in the future. For instructions see: http://journals.plos.org/plosone/s/submission-guidelines#loc-laboratory-protocols

We look forward to receiving your revised manuscript.

Kind regards,

Shinya Tsuzuki, MD, MSc

Academic Editor

PLOS ONE

Reviewers' comments:

Reviewer's Responses to Questions

**Comments to the Author**

1. If the authors have adequately addressed your comments raised in a previous round of review and you feel that this manuscript is now acceptable for publication, you may indicate that here to bypass the “Comments to the Author” section, enter your conflict of interest statement in the “Confidential to Editor” section, and submit your "Accept" recommendation.

Reviewer #1: All comments have been addressed

Reviewer #2: (No Response)

2. Is the manuscript technically sound, and do the data support the conclusions?

Reviewer #1: Yes

Reviewer #2: Yes

3. Has the statistical analysis been performed appropriately and rigorously? 

Reviewer #1: Yes

Reviewer #2: Yes

4. Have the authors made all data underlying the findings in their manuscript fully available?

Reviewer #1: Yes

Reviewer #2: Yes

5. Is the manuscript presented in an intelligible fashion and written in standard English?

Reviewer #1: No

Reviewer #2: Yes

6. Review Comments to the Author

Reviewer #1: Thank you for addressing my comments. I agree with your comments on the difference between the PHE study and yours. A key difference between the two was the impact of vaccination of English school children, making any vaccination of other population groups less impactful. I can now recommend this manuscript for publication.

Reviewer #2: The authors have adequately responded to my previous comments except for one thing. They now added a new subsection "Analyses" to describe perspectives following my previous comment. Please insert references for the definition of the two perspectives they provided as I wonder if societal perspectives should include productivity losses.

7. PLOS authors have the option to publish the peer review history of their article (what does this mean?). If published, this will include your full peer review and any attached files.

Reviewer #1: No

Reviewer #2: No

---

## [Author Response · Author response to Decision Letter 1]

4 May 2020

Following your recommendation, we have added the reference, specified hereafter, to the manuscript (ref. 51). This reference details the recommendations for economic evaluations of health technologies in Spain. We also now mention in the text the scope of indirect costs.

---

## [Editor Report · Decision Letter 2]

7 May 2020

From Trivalent to Quadrivalent Influenza Vaccines: Public Health and Economic Burden for Different Immunization Strategies in Spain

PONE-D-20-03263R2

Dear Dr. Crepey,

We are pleased to inform you that your manuscript has been judged scientifically suitable for publication and will be formally accepted for publication once it complies with all outstanding technical requirements.

With kind regards,

Shinya Tsuzuki, MD, MSc

Academic Editor

PLOS ONE
---

## [Editor Report · Acceptance letter]

11 May 2020

PONE-D-20-03263R2 

From Trivalent to Quadrivalent Influenza Vaccines: Public Health and Economic Burden for Different Immunization Strategies in Spain 

Dear Dr. Crépey:

I am pleased to inform you that your manuscript has been deemed suitable for publication in PLOS ONE. Congratulations! Your manuscript is now with our production department. 

With kind regards,

on behalf of

Dr. Shinya Tsuzuki 

Academic Editor

PLOS ONE